# A Combination of Ilizarov Frame, Externalized Locking Plate and Tibia Bridging for an Adult with Large Tibial Defect and Severe Varus Deformity Due to Chronic Osteomyelitis in Childhood: A Case Report

**DOI:** 10.3390/medicina59020262

**Published:** 2023-01-29

**Authors:** Pan Hong, Yuhong Ding, Ruijing Xu, Saroj Rai, Ruikang Liu, Jin Li

**Affiliations:** 1Department of Orthopaedic Surgery, Union Hospital, Tongji Medical College, Huazhong University of Science and Technology, Wuhan 430022, China; 2Second Clinical School, Tongji Medical College, Huazhong University of Science and Technology, Wuhan 430030, China; 3Department of Orthopedics, Al Ahalia Hospital, Mussafah, Abu Dhabi P.O. Box 2419, United Arab Emirates; 4Department of Endocrinology, Union Hospital, Tongji Medical College, Huazhong University of Science and Technology, Wuhan 430022, China

**Keywords:** bone defect, Ilizarov frame, locking plate, chronic osteomyelitis, varus deformity

## Abstract

*Background*: Various techniques have been reported to treat large, segmental tibial defects, such as autogenous bone graft, vascularized free fibula transfer and bone transport. We present a case of a 24-year-old male with a 17-year history of chronic osteomyelitis with obvious lower limb length discrepancy and severe varus deformity of the tibia secondary to osteomyelitis in childhood. *Aim*: The aim of this work is to provide an alternative choice for treating patients in developing countries with severe lower limb deformity caused by chronic osteomyelitis. *Case Presentations*: Without surgical intervention for a prolonged period of time, the patient was admitted in our institute for corrective surgery. Corrective surgery consisted of three stages: lengthening with Ilizarov frame, removal of Ilizarov frame and fixation with externalized locking plate, and removal of externalized locking plate. Tibia bridging was achieved at the distal and proximal junction. The range of motion (ROM) of the knee joint was nearly normal, but the stiffness of the ankle joint was noticeable. The remaining leg discrepancy of 0.1 cm required no application of a shoe lift. Moreover, the patient could engage in daily activities without noted limping. *Conclusions*: Distraction–compression osteogenesis using the Ilizarov apparatus is a powerful tool to lengthen the shortened long bone and adjust the deformity of the lower limbs. Externalized locking plates provide an alternative to the traditional bulky external fixator, as its low profile makes it more acceptable to patients without compromising axial and torsional stiffness. In all, a combination of Ilizarov frame, externalized locking plate and tibia bridging is an alternative for patients in similar conditions.

## 1. Introduction

Chronic osteomyelitis usually results from poorly treated or untreated acute osteomyelitis, open fractures and severe surgical complications [1]. It is seen more frequently in developing countries [2]. Several factors contribute to this situation, including virulence of bacteria, delayed presentation, poor nutritional and immune status and low social-economic status with limited access to antimicrobial agents [3].

Without proper treatment, osteomyelitis in childhood might lead to severe complications, including growth disturbance, severe leg length discrepancy (LLD) and varus or valgus malalignment [4]. The treatment strategy could be quite challenging if the residual deformity remains untreated till adulthood. With underlying osteomyelitis, staged treatment might be appropriate. As for the massive segmental bone defect in the tibia, various techniques have been proposed, including simple autogenous bone graft, allograft reconstruction, vascularized free fibula transfer, Masquelet technique and bone transport [5,6,7,8,9,10].

Here, we present a case of a 24-year-old male with residual deformity from chronic osteomyelitis in childhood. Severe LLD (cm), varus deformity (35 degrees), recurvatum deformity (33 degrees), internal rotational deformity (12 degrees), segmental tibial defect and hypertrophy of fibula was noteworthy. The patient was limping at presentation. Unlike the typical case of segmental defect of the tibia with the normal fibula, the long medical history and compensation mechanism resulted in a massive defect in the middle of the tibia, the hypertrophy of the fibula and the spontaneous bony fusion of proximal tibiofibular joint and distal tibiofibular syndesmosis. The compensatory hypertrophic fibula connected the tibial stump proximally as a *bridge.* Therefore, we distracted the fibula and strengthened the fusion between the stumps and *bridge*. We hope to provide an alternative choice for treating patients in developing countries with severe lower limb deformity caused by chronic osteomyelitis.

## 2. Case Presentation

All procedures performed in this study were in accordance with the ethical standards of the national research committee and the 1964 Helsinki declaration and its later amendments or comparable ethical standards. The patient provided informed consent for publication of this case.

A 24-year-old male was diagnosed with suppurative osteomyelitis and treated at a local hospital 17 years ago. The treatment details were unknown, and the osteomyelitis did not relapse in the past decade. At the outpatient clinic visit, he demonstrated limping with obvious lower limb length discrepancy (14 cm) and severe varus deformity of the tibia. Pelvic tilt was noticeable, and pseudoarthrosis of the tibia was formed. His ankle function was assessed by American Orthopedics Foot and Ankle Score (AOFAS) in the outpatient clinic and resulted in a score of 55 points (pain: 20, function: 25, alignment: 10) [11]. The muscle strength of the ankle assessed by the Lovett scale was level III in all directions. No active infection was detected.

## 3. Surgical Technique

The treatment strategy consisted of three staged surgeries.

Stage 1. Correction of the varus deformity and the shortening: In October 2016, the mid-section of the tibia was exposed to excise the necrotic and fibrous tissue, and synthetic bone with vancomycin was placed after copious irrigation with normal saline. Osteotomy was performed by drilling multiple holes beneath the proximal tibial fibular joint. Distal tibiofibular syndesmosis was explored, and osseous fusion was found. The Ilizarov frame was installed with 2 rings in the proximal tibia and fibula and 2 rings in the distal tibia. Seven days after the operation, the adjustment of the Ilizarov frame began (see Figure 1 and Figure 2). Distraction was performed as 1 mm per day. After 4 months of distractions, the operated leg was lengthened from 31 cm to 45 cm (total distraction index: 1.17 mm/d).

Stage 2. Removal of Ilizarov apparatus and fixation of the tibia with externalized locking plate: After the length of the operated leg was adjusted to the same as the contralateral leg and mineralization was continued for eight months, the bulky Ilizarov apparatus was removed in October 2017. However, the union between the proximal tibia and fibula was not fully consolidated based on the radiological manifestation. In order to maintain the axis and stability of the lower extremity, a 5.0 mm locking plate was used to fixate the tibia and fibula. Autologous bone was harvested at the ipsilateral iliac crest and placed at the proximal interface between the tibia and fibula (see Figure 3).

Stage 3. Removal of the externalized locking plate: The patient was encouraged to walk with the support of two canes for three months after placing the locking plate. After that, the patient was able to walk freely with the externalized locking plate. Follow-up was carried out at regular intervals (once a month for the first three months and once every three months subsequently) on an outpatient basis. The locking plate was removed after 15 months (in January 2019), and the leg was immobilized in a slab for one month before full weight bearing (see Figure 4).

At stage 1, the necrotic and fibrous tissue was sent for pathohistological examination, and chronic osteomyelitis was confirmed. Intravenous empiric antibiotic treatment was continued for two weeks after the first surgery. Routine blood tests, including complete blood count (CBC), erythrocyte sedimentation rate (ESR) and C-reactive protein (CRP) were tested every two weeks to monitor the possible relapse of the infection. Bacterial culture from necrotic tissue reported negative for infection.

## 4. Results

The patient recovered uneventfully in the early stage of reconstructive surgery of lengthening and varus correction. The shin had been lengthened by 14 cm by the help of Ilizarov frame without knee or ankle spanning. In the later follow-up, the patient was able to walk with the Ilizarov apparatus, but it was quite inconvenient for him to wear such a bulky device. At the 12th-month follow-up visit, the distraction site displayed normal bone regeneration without complete consolidation. Therefore, a 5 mm locking plate was used to replace the Ilizarov frame to stabilize the tibia. Bone grafting was performed at the proximal interface of the tibia and fibula. The patient was braced for one month before weight-bearing exercise. The patient could walk freely with the support of an externalized locking plate, and he was satisfied with the low-profile external fixator. No wound infection, fracture of the tibia and fibula, or implant failure occurred during the 15-month follow-up. At the last follow-up visit before removing the plate, the ROM of the knee joint was nearly normal (135 degrees of flexion and 5 degrees of extension), but the stiffness of the ankle joint was noticeable. Assessment of the AOFAS score resulted in a score of 85 points (pain: 35, function: 40, alignment: 10). The muscle strength of the ankle assessed by the Lovett scale was level IV. The remaining leg length discrepancy of 0.1 cm required no application of a shoe lift. In addition, after removing the plate, the patient could engage in daily activities without limping. Moreover, the patient received empiric oral antibiotic therapy twice due to pin tract infections (PTI). However, the pin was fixed firmly without loosening, and no bone destruction and osteomyelitis was demonstrated by X-ray. Therefore, the olive wire was retained, and the skin and pin disinfection were performed daily. Fortunately, the PTI was controlled uneventfully at both times. During the last outpatient visit, the patient demonstrated a full recovery. In the recent phone call follow-up, the patient reported no adverse event and was satisfied with the status quo. Quality of life was assessed by the SF-36 scale during clinical visits, and the patient demonstrated a satisfactory activity of daily living in the last phone call follow-up. No serious complications (severe contracture or dislocation, etc.) happened during the treatment.

## 5. Discussion

This is a case of a young patient with segmental tibial bone defect and significant varus deformity of the tibia caused by chronic osteomyelitis (no relapse over the last ten years). The key is the correction of bone defect with misalignment in the lower extremity.

For patients with large tibial defect, there are three available choices: (1) amputation followed by prosthetics: Amputation could avoid the relapse of chronic osteomyelitis. However, the stigma after amputation could be prominent, and the young adult rejected this treatment decidedly. (2) Limb salvage surgery with custom-made implants to reconstruct the joint: A custom-made prosthetic from 3-D printing technology is a feasible choice for patients with severely destructed anatomical structures [12], but the severe LLD of this patient makes a custom-made implant unsuitable. (3) Limb salvage surgery with bone lengthening and tibial bridging: Autograft is not a feasible choice when the bone defect is more than 6–8 cm. Various apparatuses have been reported for bone lengthening, including Ilizarov frame, Taylor spatial frame (TSF) or Hexapod, and PRECICE [6,7,8,9].

PRECICE is an intramedullary limb lengthening system approved by the FDA; excellent outcomes with less pain and lower complication rates have been reported in over 250 cases [7]. However, this technique is not feasible for patients with abnormal intramedullary canal, as in this case. Additionally, although TSF and Hexapod are powerful computer-assisted deformity corrective tools, TSF is expensive and Hexapod is not available in China [8,9]. Moreover, for significant varus deformity with severe LLD, acute correction of angular deformity requires secondary surgery for lengthening. Therefore, we adopted the Ilizarov frame with spatial corrective potential, which provides relatively satisfactory effects and lower financial burden.

When the desired length and alignment is achieved, the patient wanted the bulky Ilizarov apparatus removed. There were several choices after hardware removal: (1) plaster support or brace, (2) rigid tibial nail, (3) plating, (4) externalized plating. Casting or bracing would restrict the weight-bearing exercises and might result in joint stiffness. Nailing is not suitable since the hypertrophic fibula does not have a normal intramedullary canal. The skin coverage and pin tract make conventional plating not applicable. Therefore, externalized plating was adopted in our study.

There were several details for this patient: a unilateral external fixator was not applicable due to delayed weight bearing and limited angular correction potential. Schanz screws in a one-sided external fixator could not provide sufficient purchase and accomplish massive lengthening. Therefore, Ilizarov frame was adopted. Moreover, for this patient, the compensation mechanism resulted in the osseous union between the tibia and fibula at the proximal and distal junction, and the hypertrophy of the fibula was so strong that it could withstand the body weight. Therefore, the compensatory fibula could work as a *bridge* to connect the tibia at both ends. With regard to the possible residual inflammation in the shaft of the tibia, we chose to remove the necrotic section of the tibia and elongate the hypertrophied fibula to correct the significant bone defect. Moreover, since the proximal and distal junction between the tibia and fibula was fused before corrective surgery, centralization of the fibula was not performed for better axial alignment. However, without the end-to-end union, the patient in our study demonstrated a normal gait and satisfactory daily life quality with tibial bridging.

As for the lengthening process, the recommended lengthening magnitude was 5–8 cm in long bones [13]. When the bone defect is substantial, the time to achieve the desired length can be extremely long. The lengthening process could be fraught with complications, such as PTI, neurovascular injury, muscle weakness, axial malalignment and nonunion or delayed union. Fortunately, no major complications such as neurovascular injury, infection, muscular damage, deep venous thrombosis (DVT) and re-fractures occurred. Serious complications such as severe contracture or dislocation were avoided during the treatment. However, the patient complained of intermittent PTI as well as the inconvenience of the bulky device in daily activities. The lengthening process was uneventful, but the mineralization and consolidation were insufficient at the time of Ilizarov apparatus removal. Concerning biomechanics, the stability of traditional plate fixation was worse than the locking plate, and the locking plate provided good angle stability [14]. Therefore, a 5 mm externalized locking plate was adopted after removing the Ilizarov apparatus. One month of immobilization of the long leg slab was implemented after the removal of the Ilizarov apparatus. Partial weight bearing was allowed after the removal of the slab, and progression to full weight bearing was initiated with discretion. Three months after the removal of the Ilizarov apparatus, the patient was able to walk freely with the externalized locking compression plate (LCP).

Postoperative complications have also been taken into account. PTI is common, but it is controllable with timely administration of oral antibiotics [15]. Rogers et al. reported that PTI occurred in 10% to 20% of cases of Ilizarov external fixators [16]. Moreover, they demonstrated the application of gentle compression dressings and reducing hyperglycemia and tourniquet time could reduce the rate of PTI. In addition, the bulky apparatus reduces patients’ compliance, and wearing this device may decrease the quality of life and psychological happiness, which has been observed in adolescents [17]. Therefore, we removed the Ilizarov frame and changed it into a low-profile externalized plate. However, neurovascular injury, muscle weakness, axial malalignment and nonunion or delayed union are still common complications in massive lengthening, and discretion is required for patients with this condition [18,19,20,21].

There have been multiple cases reporting the application of externalized LCP in various regions of the body [22]. It has been shown that the main factors affecting the stiffness of LCP include working length, number of screws, distance from the plate to the bone and length of the plates [23]. Thirty millimeters is the upper bound of bone–plate distance to keep fixation stable in distal tibia fractures [24,25]. The increased diameters of the screws and the dimensions of the plate significantly enhance torsional rigidity but contribute little to compression stiffness. It is reported that externalized LCP could bear three times the body weight of an average 70 kg adult on axial loading only [26]. Thus, a 5.0 mm distal femoral locking plate was used in this patient.

Similarly, multiple cases have been reported regarding the Ilizarov frame on massive tibia lengthening. Kawoosa et al. reported a case of performing 20 cm lengthening and complex deformity correction on a centralized fibula using the Ilizarov technique with an excellent result [27]. They spent 17 months to complete the process of lengthening and consolidation at a healing index of 0.85 months/cm. Alkenani et al. also reported a case of right Gustilo IIIA segmental open tibia fracture with bone loss and other severe injuries [28]. Although the tibial defect was 14.5 cm, the patient was then admitted for Ilizarov application six months after the accident, and full limb length was restored. These cases corroborated that the Ilizarov frame was a positive alternative for patients with massive tibia lengthening.

To the best of our knowledge, no similar case with massive tibial defect and severe varus deformity due to chronic osteomyelitis was treated with Ilizarov frame, externalized plating and tibial bridging has been reported. Staged surgery is safe and relatively low-cost. The merits of our choices have been elucidated in the aforementioned context. Tibial bridging with successful fusion should be the touchstone for our staged treatment.

There also existed several limitations in our study. Firstly, meticulous care of a professional team with profound experience is usually required during the lengthening process. Moreover, our study was a case report rather than a cohort study, and externalized locking plate was unconventional. Moreover, the TSF frame might have demonstrated better outcomes than the Ilizarov frame, but TSF is more expensive than the Ilizarov frame. In addition, centralized fibula might be a better choice than tibial bridging.

## 6. Conclusions

Our study focuses on a patient with severe lower extremity deformity from chronic osteomyelitis in childhood. The Ilizarov apparatus was a powerful tool to lengthen the shortened long bone and adjust the deformity of the limbs in distraction–compression osteogenesis. Externalized locking plate provides an alternative to a traditional bulky external fixator because its low profile makes it more acceptable to patients without compromising axial and torsional stiffness. Moreover, in patients with static chronic osteomyelitis, a one-stage operation of necrotic bone resection and distraction osteogenesis is a feasible choice. In all, a combination of Ilizarov frame, externalized locking plate and tibia bridging is an alternative for patients in similar conditions.

## Figures and Tables

**Figure 1 medicina-59-00262-f001:**
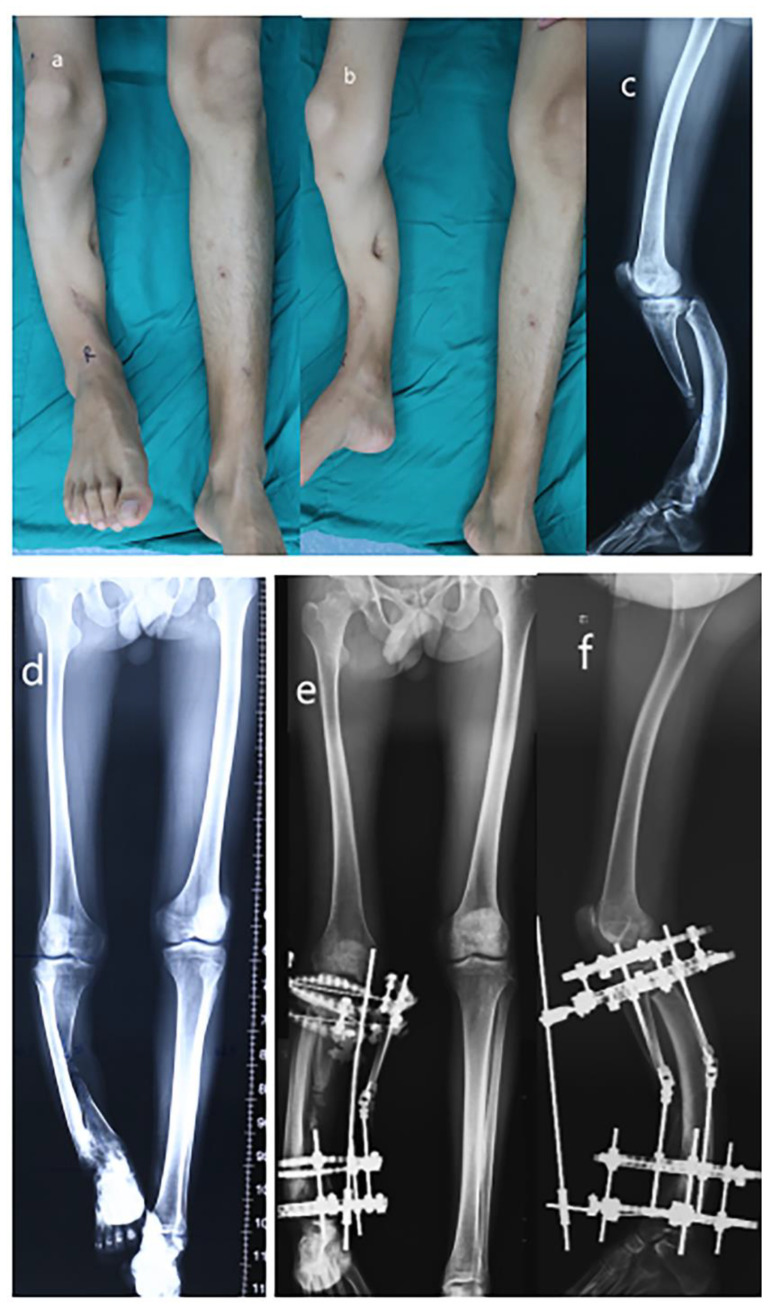
Radiograph of a 24-year-old male of severe limb length discrepancy and tibia varus. (**a**) Appearance (AP view); (**b**) Appearance (lateral view); (**c**) Preoperative later view radiograph; (**d**) Preoperative AP view radiograph; (**e**) Postoperative AP view radiograph; (**f**) Postoperative lateral view radiograph.

**Figure 2 medicina-59-00262-f002:**
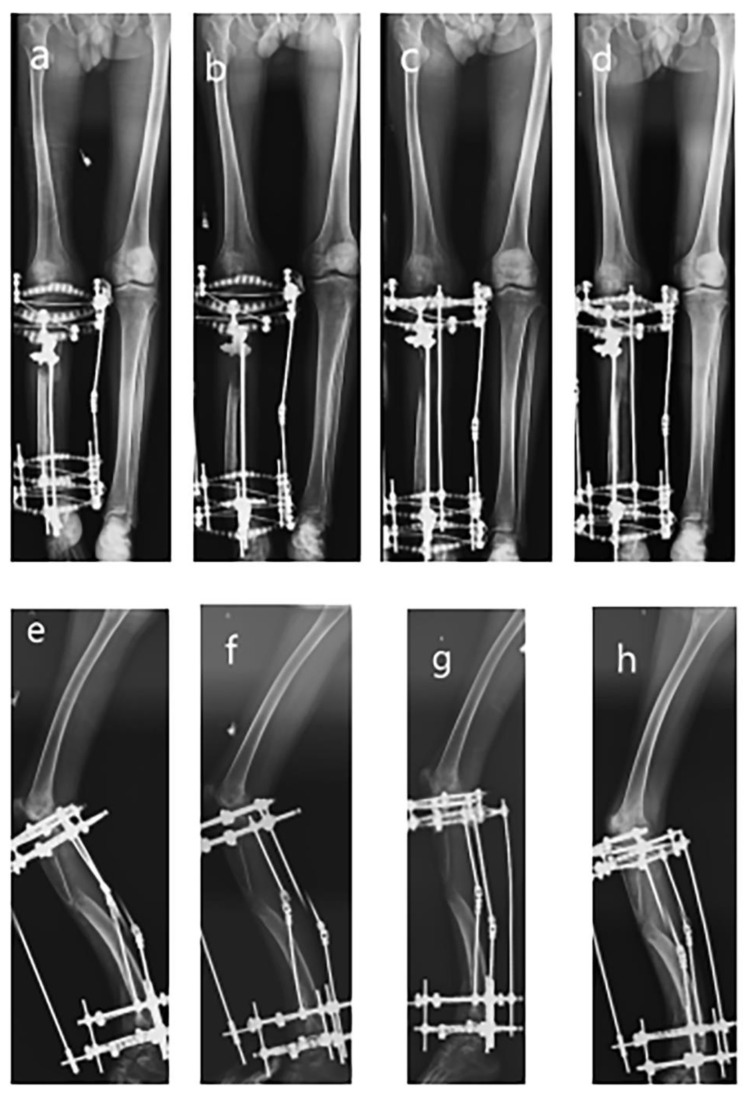
Series radiographs of the lower extremity after Ilizarov apparatus instalment. (**a**) AP view radiograph of 5th-month follow-up; (**b**) AP view radiograph of 8th-month follow-up; (**c**) AP view radiograph of 9th-month follow-up; (**d**) Lateral view radiograph of 10th-month follow-up; (**e**) Lateral view radiograph of 5th-month follow-up; (**f**) Lateral view radiograph of 8th-month follow-up; (**g**) Lateral view radiograph of 9th-month follow-up; (**h**) Lateral view radiograph of 10th-month follow-up.

**Figure 3 medicina-59-00262-f003:**
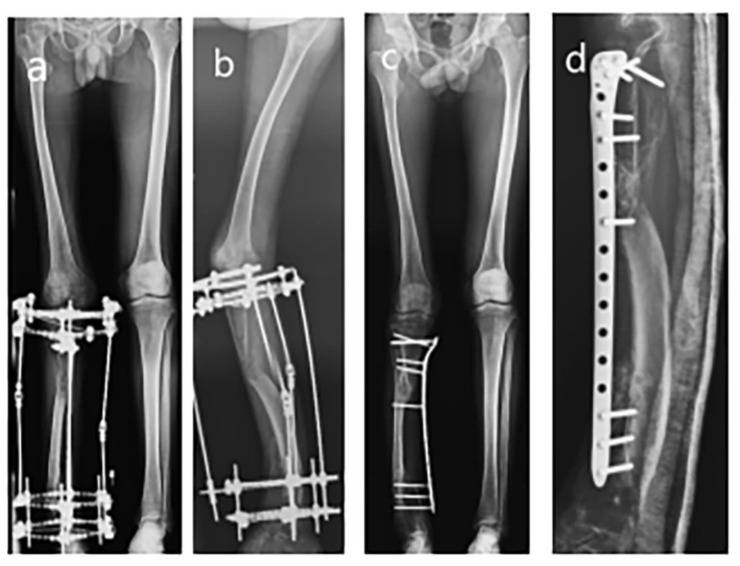
Removal of Ilizarov apparatus and externalized LCP. (**a**) AP view radiograph before removal of Ilizarov apparatus; (**b**) Lateral view radiograph before removal of Ilizarov apparatus; (**c**) AP view radiograph of externalized LCP fixation; (**d**) Lateral view radiograph of externalized LCP fixation.

**Figure 4 medicina-59-00262-f004:**
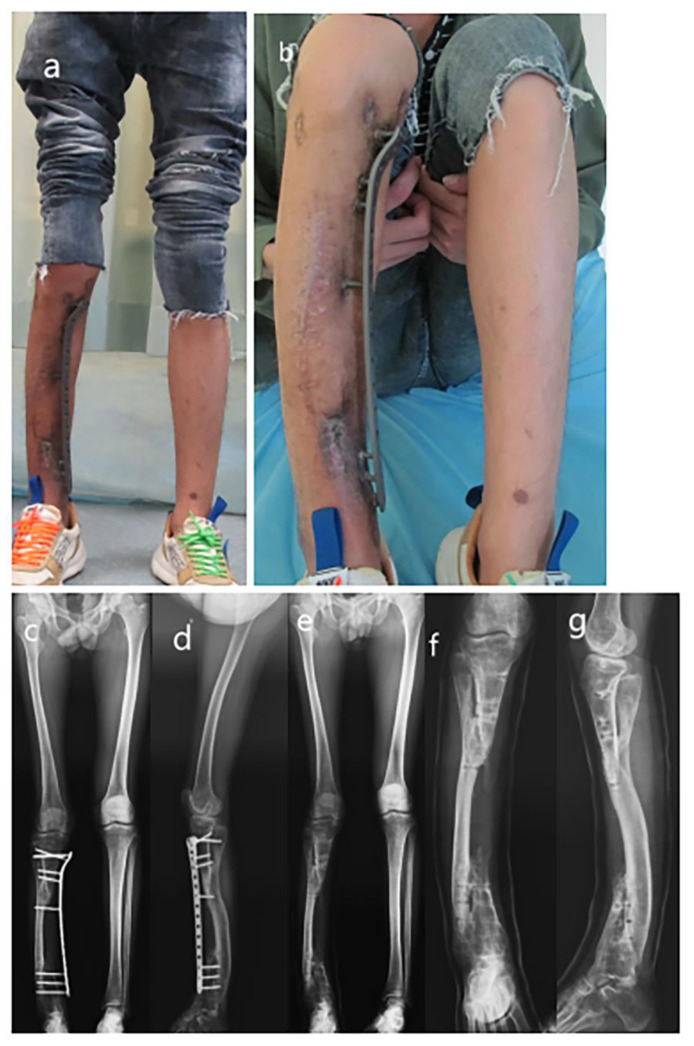
The appearance of the lower extremity and removal of the plate. (**a**) The appearance of the lower extremities; (**b**) Flexion of the knee joint; (**c**) AP view full-length radiograph of lower extremities before plate removal; (**d**) Lateral view full-length radiograph of lower extremities before plate removal; (**e**) AP view full-length radiograph of lower extremities after plate removal; (**f**) AP view radiograph of tibia–fibula after plate removal; (**g**) Lateral view radiograph of tibia–fibula after plate removal.

## Data Availability

Data sharing is not applicable to this article as no datasets were generated or analyzed during the current study.

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
