# Peer review of "A Combination of Ilizarov Frame, Externalized Locking Plate and Tibia Bridging for an Adult with Large Tibial Defect and Severe Varus Deformity Due to Chronic Osteomyelitis in Childhood: A Case Report"

_medicina, 2023, doi:10.3390/medicina59020262_

Round 1

Reviewer 1 Report

The issue proposed by the study is interesting especially because it refers to a treacherous disease, still difficult to treat and manage.

The abstract is comprehensive and exhaustive.

The methodological approach is correct and the case presentation is clearly reported.

Images and results used are well described. 

The study might be worthy of publication after a minor revision:

- A native English speaker might be useful to proofread the text to improve the grammatical content;

- Advantages and disadvantages of the technique used might be more widely described;

- Comparison with the other surgical procedures reported could be better discussed.

- In the discussion, among the various surgical techniques described, it might be exhaustive to argue about custom-made 3D-printed patient-specific spacers and to mention the following article:

Custom-Made Implants in Ankle Bone Loss: A Retrospective Assessment of Reconstruction/Arthrodesis in Sequelae of Septic Non-Union of the Tibial Pilon" 

doi: 10.3390/medicina58111641

    Author Response

    The issue proposed by the study is interesting especially because it refers to a treacherous disease, still difficult to treat and manage. The abstract is comprehensive and exhaustive. The methodological approach is correct and the case presentation is clearly reported. Images and results used are well described. 

    Response: Thanks for your comments.

     The study might be worthy of publication after a minor revision:

    - A native English speaker might be useful to proofread the text to improve the grammatical content;

    Response: Thank you. A native speaker has revised our manuscript.

    - Advantages and disadvantages of the technique used might be more widely described;

    Response: Advantages and disadvantage have revised as request.

    - Comparison with the other surgical procedures reported could be better discussed. In the discussion, among the various surgical techniques described, it might be exhaustive to argue about custom-made 3D-printed patient-specific spacers and to mention the following article:

    “Custom-Made Implants in Ankle Bone Loss: A Retrospective Assessment of Reconstruction/Arthrodesis in Sequelae of Septic Non-Union of the Tibial Pilon" 

    doi: 10.3390/medicina58111641

    Response: We have added a whole paragraph in Discussion part to describe the Comparison with the other surgical procedures, and the article above is included (see reference 12).

    Reviewer 2 Report

    Thank you very much for giving me the opportunity to review this article. I would like to congratulate the authors for combining different methods of treatment to obtain a fairly good result of lengthening and deformity correction.

    I have however doubts that the content of this article, and the way this article is presented, are sufficient enough.

    Here are my concerns/comments:

    Major concerns:

    -This is a single case presentation – this, according to EBM standards, is Level V evidence, and should not be therefore considered as the impacted publication, as long as it brings something completely new to the medical practice. In this case, methods of treatment used here have been utilized before in different configurations. Therefore, the impact of this presentation is not sufficient enough to be considered worth publishing in the IF journal. 

    - Authors describe this case as a combined method of lengthening and cross-union with both Ilizarov frame and Ex-fix locking plate. In the case description they state that the distraction and correction osteotomy was done below tib-fib synostosis. It looks to me that it was mainly done as the fibula osteotomy and the bony fusions between tibia and fibula proximally and distally were utilized as “tibia bridging”. I understand, therefore, that the cross union which authors meant in their article’s title is actually what they reinforced in the proximal tib-fib fusion during the second stage of treatment. This has to be emphasized in both introduction and discussion. 

    -Also, my feeling is that the authors have used what patient’s biology brought – the hypertrophied fibula which bridged the tibial defect. That is a very smart way of using it and I would like to congratulate them for what they have done. However, in the article’s introduction and discussion they compare bone transport or bone harvesting to what they have done – and it looks to me a bit like comparing apples and pears. Therefore, the whole introduction and discussion should be re-edited to make clear what were the rationale of the treatment, and how this method could work as an alternative to the other methods of treatment of large bone defects.

    -Due to poor quality of figures, it is impossible to fairly assess the obtained results.

     Minor comments:

    1. Introduction :

    - what is the aim of this case presentation? Please add the aim to this part.

     2.Case presentation:

    “The muscle strength of the ankle assessed by the Lovett scale was level III.” – all directions? Please, elaborate on that. 

     3. Surgical technique:

    -“Osteotomy was carried out by multiple drill holes beneath the proximal tibial fibular joint for distraction.” -please rephrase.

    -“ After 4 months of distractions, the operated leg was lengthened from 31 cm to 45 cm (total distraction index: 0.452).” – That means the leg was lengthened 14 cm in around 120 days. Distraction index is usually interpreted as the amount of lengthening in mm per day. This does not add up with either of numbers given here. 

    -“On the radiograph, the union between the proximal tibia and fibula was not fully consolidated.” Please rephrase that sentence. 

    4. Discussion:

    “Bone transport is an alternative for post-traumatic and post- infectious growth disturbance. In all, Ilizarov frame, Taylor spatial frame (TSF), PRECICE, and Hexapod have been reported for substantial bone defect [6-9].” – TSF and Hexapod should be combined in one category. 

    -       Please elaborate on the fact that the shin has been legntehened by 14 cm in the frame without knee or anke spanning and without complications like severe contractures or dislocations.

    5. Conclusion

    Conclusion should be written based on the research aims, which were missing in this case. Please, rewrite. 

    6. Language: I feel that the article needs to be edited by native English speaker .

    Author Response

    Thank you very much for giving me the opportunity to review this article. I would like to congratulate the authors for combining different methods of treatment to obtain a fairly good result of lengthening and deformity correction. I have however doubts that the content of this article, and the way this article is presented, are sufficient enough.

    Response: Thanks for your comments.

    Here are my concerns/comments:

     Major concerns:

    -This is a single case presentation – this, according to EBM standards, is Level V evidence, and should not be therefore considered as the impacted publication, as long as it brings something completely new to the medical practice. In this case, methods of treatment used here have been utilized before in different configurations. Therefore, the impact of this presentation is not sufficient enough to be considered worth publishing in the IF journal. 

    Response: To the best of our knowledge, no similar case with massive tibial defect and severe varus deformity due to chronic osteomyelitis was treated with Ilizarov frame, externalized plating and tibial bridging has been reported. Despite Ilizarov frame, externalized plating had been reported separately in different studies.

    - Authors describe this case as a combined method of lengthening and cross-union with both Ilizarov frame and Ex-fix locking plate. In the case description they state that the distraction and correction osteotomy was done below tib-fib synostosis. It looks to me that it was mainly done as the fibula osteotomy and the bony fusions between tibia and fibula proximally and distally were utilized as “tibia bridging”. I understand, therefore, that the cross union which authors meant in their article’s title is actually what they reinforced in the proximal tib-fib fusion during the second stage of treatment. This has to be emphasized in both introduction and discussion. 

    Response: Indeed, we have changed all description of “cross-union” into “tibia bridging”, and added description of tibia bridging in the Introduction and Discussion part.

    -Also, my feeling is that the authors have used what patient’s biology brought – the hypertrophied fibula which bridged the tibial defect. That is a very smart way of using it and I would like to congratulate them for what they have done. However, in the article’s introduction and discussion they compare bone transport or bone harvesting to what they have done – and it looks to me a bit like comparing apples and pears. Therefore, the whole introduction and discussion should be re-edited to make clear what were the rationale of the treatment, and how this method could work as an alternative to the other methods of treatment of large bone defects.

    Response: Thanks for your valuable advice. We have deleted part of manuscript as request.

    -Due to poor quality of figures, it is impossible to fairly assess the obtained results.

    Response: Sorry for the poor quality of figures. We have submitted the original figure this time.

    Minor comments:

    1. Introduction :

    - what is the aim of this case presentation? Please add the aim to this part.

    Response: We have added the aim of this case presentation in abstract and introduction part

    2.Case presentation:

    m“The muscle strength of the ankle assessed by the Lovett scale was level III.” – all directions? Please, elaborate on that. 

    Response: Indeed, The muscle strength of the ankle assessed by the Lovett scale was level â…¢ in all direction. We have revised the manuscript accordingly.

    1. Surgical technique:

    -“Osteotomy was performed by multiple drill holes beneath the proximal tibial fibular joint for distraction.” -please rephrase.

    -“Based on the radiological manifestation, the union between the proximal tibia and fibula was not fully consolidated.” Please rephrase that sentence. 

    Response: Thanks for you advice, we have revised these two sentences.

    -“ After 4 months of distractions, the operated leg was lengthened from 31 cm to 45 cm (total distraction index: 0.452).” – That means the leg was lengthened 14 cm in around 120 days. Distraction index is usually interpreted as the amount of lengthening in mm per day. This does not add up with either of numbers given here. 

    Response: Thank you, we have revised the total distraction index into 1.17mm/d

    1. Discussion:

    “Bone transport is an alternative for post-traumatic and post- infectious growth disturbance. In all, Ilizarov frame, Taylor spatial frame (TSF), PRECICE, and Hexapod have been reported for substantial bone defect [6-9].” – TSF and Hexapod should be combined in one category. 

    - Please elaborate on the fact that the shin has been legntehened by 14 cm in the frame without knee or anke spanning and without complications like severe contractures or dislocations.

    Response: Thanks for you advice, we have revised these two sentences in the Discussion part accordingly.

    1. Conclusion

    Conclusion should be written based on the research aims, which were missing in this case. Please, rewrite. 

    Response: Thank you. We have revised the conclusion as request.

    1. Language: I feel that the article needs to be edited by native English speaker .

    Response: Thank you. A native speaker has revised our manuscript.